# A Cap-Optimized mRNA Encoding Multiepitope Antigen ESAT6 Induces Robust Cellular and Humoral Immune Responses Against *Mycobacterium tuberculosis*

**DOI:** 10.3390/vaccines12111267

**Published:** 2024-11-09

**Authors:** Alena Kozlova, Ildus Pateev, Galina Shepelkova, Olga Vasileva, Natalia Zakharova, Vladimir Yeremeev, Roman Ivanov, Vasiliy Reshetnikov

**Affiliations:** 1Translational Medicine Research Center, Sirius University of Science and Technology, Sochi 354340, Russia; kozlova.av@talantiuspeh.ru (A.K.); pateev.ii@talantiuspeh.ru (I.P.); vasileva.oo@talantiuspeh.ru (O.V.); zaharova.na@talantiuspeh.ru (N.Z.); ivanov.ra@talantiuspeh.ru (R.I.); 2Central Tuberculosis Research Institute, Moscow 107564, Russia; g.shepelkova@ctri.ru (G.S.); yeremeev56@mail.ru (V.Y.)

**Keywords:** RNA vaccine, cap analog, tuberculosis, adaptive immunity, protective immunity

## Abstract

**Background/Objectives.** Tuberculosis is a deadly bacterial disease and the second most common cause of death from monoinfectious diseases worldwide. Comprehensive measures taken by health authorities in various countries in recent decades have saved tens of millions of lives, but the number of new cases of this infection has been steadily increasing in the last few years and already exceeds 10 million new cases annually. The development of new vaccines against tuberculosis is a priority area in the prevention of new cases of the disease. mRNA vaccines have already shown high efficacy against COVID-19 and other viral infections and can currently be considered a promising field of antituberculosis vaccination. In our previous study, we assessed the immunogenicity and protective activity of several types of antituberculosis mRNA vaccines with different 5′ untranslated regions, but the efficacy of these vaccines was either comparable with or lower than that of BCG. **Methods.** Here, we conducted a comprehensive experiment to investigate the effects of cotranscriptional capping conditions and of cap structure on the magnitude of the mRNAs’ translation in HEK293T and DC2.4 cells. The most effective cap version was used to create an antituberculosis mRNA vaccine called mEpitope-ESAT6. **Results and Conclusions**. We compared immunogenicity and protective activity between mEpitope-ESAT6 and BCG and found that the vaccine with the new cap type is more immunogenic than BCG. Nonetheless, the increased immunogenicity did not enhance vaccine-induced protection. Thus, the incorporation of different cap analogs into mRNA allows to modulate the efficacy of mRNA vaccines.

## 1. Introduction

According to a report from the World Health Organization (WHO), the number of new cases of tuberculosis has been steadily increasing during the past 3 years—in 2020, 10 million; in 2021, 10.3 million; and in 2022, 10.6 million—and is outpacing the growth rate of the world’s population. At the same time, the mortality rate remains high: ~1.5 million cases per year. The observed setback in the fight against tuberculosis in recent years is partly due to the excessive burden on the healthcare system during the COVID-19 pandemic [1].

The most effective way to prevent tuberculosis is vaccination. Widespread vaccination with BCG in childhood has shown strong protection from tuberculosis in children. In adults, however, BCG does not provide adequate protection against the most common type of tuberculosis: pulmonary tuberculosis [2]. Furthermore, revaccination with BCG in adulthood is not recommended by the WHO owing to possible adverse effects [3]. In this regard, the development of new vaccines against tuberculosis is an urgent and relevant task. At present, 17 vaccines against tuberculosis are at various stages of clinical trials [4]. All the vaccines under development can be categorized into the following types: vector vaccines (based on viruses and bacteria), subunit vaccines (based on several recombinant proteins), live attenuated vaccines, whole-cell vaccines, and mRNA vaccines. The antigens used for these vaccines (except for live attenuated vaccines and whole-cell vaccines) include both well-characterized proteins (Ag85, ESAT6, and CFP-10) and those identified relatively recently but showing high potential in in vivo experiments [4,5].

Among the vaccines under development, third-generation mRNA-based vaccines are especially interesting. The arsenal of mRNA vaccines includes a wide range of tools for modulating an immune response [6]. Aside from possible use of classic adjuvants, genetic adjuvants can be employed in this context and have shown good effectiveness [7]. In addition, this platform enables the production of multiepitope mRNAs encoding individual desired epitopes. This approach proved to be effective in our recent work on the preparation of mRNA vaccines against tuberculosis [8].

mRNA vaccines have shown high efficacy and acceptable safety in the fight against the COVID-19 pandemic. Nevertheless, their success as antiviral vaccines does not yet guarantee success against bacterial pathogens because the structure of bacteria and their mechanisms of evasion of an immune response are more complicated. The advent of effective delivery systems based on lipid nanoparticles (LNPs), continuous improvement of in vitro transcription technology, the discovery of modified nucleotides, and the emergence of alternative mRNA platforms based on circular and self-amplifying mRNAs inspire hope for future advancements in mRNA vaccines against tuberculosis.

One of the stages of mRNA technology optimization is the selection of optimal structural elements of mRNA: a 5′ cap, 5′ untranslated region (5′UTR), open reading frame (ORF), 3′ untranslated region (3′UTR), and poly(A) tail [9]. In our earlier report, we demonstrated that varying the 5′UTR can significantly increase the effectiveness of an mRNA vaccine against tuberculosis [10]. Another important component of mRNA is the 5′ cap because its presence is necessary for stable and efficient translation.

The 5’ cap is a modified *N*7-methylguanosine connected via a 5′–5′ triphosphate bridge to the first nucleotide of a transcript (m^7^GpppN). This structure is called cap 0 and in the cell, it plays a key role in several important biological functions. The presence of a cap facilitates transcription initiation, polyadenylation, and splicing [11,12,13,14]. The 5′ cap protects mRNA from degradation by 5′ exoribonucleases and promotes cap-dependent translation (protein synthesis) [14,15,16]. Most often, cap 0 undergoes additional methylation at the 2′-O-ribose position of the first nucleotide in mRNA. This structure is termed cap 1 (m^7^GpppNm) and additionally serves as a marker for the innate immune system by helping to distinguish viral RNAs from cellular ones [17]. Cytoplasmic functions of the cap are especially important when heterologous mRNAs are used in therapeutics.

The efficiency of protein synthesis from mRNA is largely determined by the structure of the 5′ cap. In eukaryotic cells, the main mode of translation initiation is the cap-dependent scanning mechanism, the rate of which depends on the binding of initiation factor eIF4E to the 5′ cap [16,18]. The duration of mRNA existence (or its stability) in the cell also substantially depends on cap structure [14,15]. This characteristic is determined by two processes: the resistance of a cap to decapping (this resistance allows mRNA to avoid degradation by 5′ exoribonucleases for a longer period) and weaker affinity for innate immune complexes (which allows mRNA to avoid degradation in this pathway). A 5′ cap can be incorporated into an mRNA sequence either during in vitro transcription (cotranscriptional capping) or after mRNA synthesis (post-transcriptional capping) by means of recombinant capping enzymes.

In cotranscriptional capping, the incorporation of a cap analog into the strand is performed by a bacteriophage RNA polymerase (T7, T3, or SP6). The incorporation of the cap analog is a competitive reaction and depends on the proximal cap nucleotide and on the cap-to-GTP ratio. T7 polymerase can initiate transcription not only from the 3′-OH of cap analog guanosine but also from GTP, thereby resulting in the formation of uncapped RNA. To increase the number of initiation events involving a cap analog (capping efficiency), the cap/GTP ratio is usually varied.

In our work, we compared the translation efficiency of mRNAs capped with one of three analogs carrying an additional methyl group at the 3′-O position of m^7^G ribose in terms of luciferase gene expression in cell lines DC2.4 and HEK293T. In our assays, we utilized different ratios of cap analogs of different structures to GTP and assessed how these variations affect the presence of admixtures of double-stranded RNAs (dsRNAs) generated from in vitro-transcribed mRNAs. In addition, we assessed the influence of the most effective 5′ cap (judging by cell culture data) in an mRNA vaccine against *Mycobacterium tuberculosis* on the formation of adaptive and protective immunity in mice.

## 2. Materials and Methods

### 2.1. The In Vitro Experiment

The goal of the first stage of this work was to choose the best type of synthetic cap analog that ensures strong translation of a heterologous RNA. For comparison, we selected three modified cap analogs of the ARCA (anti-reverse cap analog) class; they differed in the cap’s proximal nucleotide: m_2_^7,3′-O^GpppG “ARCA”, m_2_^7,3′-O^GpppAmG “CapAG”, and m_2_^7,3′-O^GpppGmG “CapGG” (Figure 1a). To obtain mRNA samples with different degrees of capping, we varied the ratio of a cap analog to GTP in the reaction mixture. For the caps with a proximal guanosine (ARCA and CapGG), ratios of 2:1, 4:1, and 8:1 were chosen, and for the cap analog with a proximal adenosine (CapAG), the ratios were 0.4:1.0, 0.8:1.0, and 1:1. The protocol used for the synthesis of mRNA was optimized for the 4:1 ratio for ARCA and for the 0.8:1.0 ratio for CapAG. The reporter RNA encoded the sequence of firefly luciferase (LucF); for the correct incorporation of a cap analog, two plasmids differing by one nucleotide were employed as a template (Figure 1b). Nine capped mRNAs (three types for each cap analog) and two types of uncapped RNA (from each DNA template) were prepared for assays. The translation magnitude of the resulting RNAs was assessed through transfection of DC2.4 and HEK293T cell lines and via evaluation of luciferase bioluminescence intensity at 4, 24, and 48 h after the transfection (Figure 1c).

### 2.2. In Vitro Transcription and RNA Purification

All DNA constructs utilized in the work were prepared on the basis of the commercial pSmart vector (Lucigen, Middleton, WI, USA). For in vitro experiments, two constructs were used—pS-5′(GG)Mod-LucF-Mod3′_114A and pS-5′(AG)Mod-LucF-Mod3′_114A—linearized by endonuclease *Ahl*I (SibEnzyme, Moscow, Russia), which have been described previously [19]. Both contained (i) a cassette encoding the 5′UTR from an mRNA vaccine called mRNA-1273 (Moderna, Cambridge, MA, USA), (ii) the sequence of firefly luciferase acting as a reporter gene, (iii) the 3′UTR from mRNA vaccine mRNA-1273 (Moderna), and (iv) a polyA tail of 114 nucleotides. For the in vivo experiment, plasmid pS-5′TPL-mEpitope-ESAT6-Mod3′ 114A described previously [10] was employed. The resulting vectors were used to transform NEB-stable cells (New England Biolabs, Hitchin, UK), which were cultured at 30 °C and 180 rpm.

The in vitro transcription reaction (50 μL) was carried out by means of the mRNA-20 kit (Biolabmix, Novosibirsk, Russia) and incubated at 37 °C for 2 h. For uncapped RNA synthesis, the mixture consisted of 5× buffer, 10× DTT, 18 U/μL T7 polymerase, sterile RNase-free deionized water, a set of four triphosphorylated nucleotides at 3 mmol/μL (ATP, UTP, CTP, and GTP), and 1 μg of a linearized plasmid as a template. In addition, 1 U/μL RNase inhibitor RiboCare (Evrogen, Moscow, Russia) and 0.002 U/μL pyrophosphatase (New England Biolabs) were added to the mixture. To synthesize a pool of capped mRNAs, one of three synthetic cap analogs was also introduced into the reaction mixture: ARCA (cat. No. ARCA-1000; Biolabmix), CapGG (synthesized at N.F. Gamaleya National Research Center for Epidemiology and Microbiology, Moscow, Russia), or CapAG (cat. No. AGME-1000; Biolabmix) in different ratios relative to GTP (Figure 1a,b). Volumes of the reagents used and their final concentrations are given in Appendix A. After 2 h of incubation, DNase I (New England Biolabs) was added to a final concentration of 0.1 U/μL, and the incubation was continued for 20 min at 37 °C. mRNA was purified from the reaction mixture with the Agencourt RNAClean XP Kit (Beckman Coulter, Brea, CA, USA).

For the in vivo experiment, the synthesis of the mRNA carrying the mEpitope-ESAT6 sequence was carried out with the addition of the trinucleotide cap analog CapGG at a molar ratio of the cap to GTP of 4:1. The composition and sequence of resulting mRNA are given in the Appendix A.

The quality of the transcripts was checked by electrophoresis in a 6% polyacrylamide gel containing 7 M urea and by electrophoresis in an MCE-202 MultiNA system (Shimadzu, Kyoto, Japan). The concentration was determined spectrophotometrically with the help of absorption at 260 nm and fluorimetrically on a Qubit 4 instrument (Invitrogen, Waltham, MA, USA) by means of Qubit RNA High Sensitivity and Broad Range kits.

### 2.3. Detection of DsRNA by a Dot Immunobinding Assay

The amount of dsRNA in mRNA samples was assessed by a dot blot assay with antibodies to dsRNA (Antibody System, Schiltigheim, France). For this purpose, RNA (2 μL at a concentration of 250 ng/μL) was applied to a nitrocellulose membrane with a pore diameter of 0.45 μm (Bio-Rad Laboratories, Hercules, CA, USA). After the application, the membrane was dried at room temperature and treated three times with UV light at 254 nm for 1 min with an interval of 10 s on a transilluminator (Vilber Lourmat, Eberhardzell, Germany). The membrane was then placed in 5% serum albumin in TBST buffer (0.1% Tween 20, 1× TBS) for 1 h incubation to prevent nonspecific binding and was washed in TBST buffer. The membrane was then incubated with an anti-dsRNA IgG antibody (dilution 1:10,000) overnight at 4 °C with slow agitation, after which the membrane was washed again with TBST. At the next step, the membrane was incubated with an anti-mouse IgG antibody conjugated with phycoerythrin (PE) (1:20,000) (Transgen Biotech Co., Beijing, China) for 1 h at room temperature and washed with TBST. R-phycoerythrin fluorescence was detected using a ChemiDoc MP system (Bio-Rad Laboratories) via 602/50 nm fluorescence.

As a standard for calculating the concentration of the dsRNA impurity, a series of control samples of dsRNA were prepared with concentrations of 5 to 0.125 ng/μL for mEpitope-ESAT6 mRNA (Appendix A) and with concentrations of 10 to 0.125 ng/μL for the reporter mRNAs. To obtain dsRNA, plasmids containing the expression cassette either in the forward orientation (pS-5′Mod-NLuc-Mod3′ 114A) or in the reverse orientation (pS-114A_Mod5′-NLuc-3′Mod) downstream of the T7 RNA polymerase promoter were utilized as templates. The expression cassette consisted of the 5′UTR from mRNA vaccine mRNA-1273 (Moderna), the NanoLuc luciferase (NLuc) sequence as a reporter gene, the 3′UTR from mRNA-1273 vaccine mRNA (Moderna), and a poly(A) tail of 114 adenosine nucleotides. The plasmids were linearized with endonuclease *Ahl*I (SibEnzyme). Complementary RNAs were transcribed in vitro and purified by LiCl precipitation. Annealing of complementary strands was performed by heating at 95 °C for 1 min in a buffer (10 mM Tris-HCl pH 7.0 and 50 mM NaCl) with subsequent cooling at 25 °C for 1 h. After hybridization, single-stranded RNA was removed by the addition of 1 U/μg nuclease S1 (Thermo Fisher Scientific, Waltham, MA, USA) and incubation for 45 min at room temperature. After precipitation with LiCl, the precipitate of dsRNA was dissolved in ultrapure water. For the quantification of dsRNA, two independent runs with two technical replicates (for reporter mRNAs) were performed, as was one run with three technical replicates (for mEpitope-ESAT6 mRNA). Within each run, standards of dsRNA were assayed along with the experimental samples.

### 2.4. Cell Cultivation and Transfection

HEK293T cells (ATCC, CRL-3216) were cultured in the DMEM GlutaMAX-I medium (Servicebio, Wuhan, China) supplemented with 10% of fetal bovine serum (HiMedia, Mumbai, India) at 5% CO_2_ and 37 °C. Murine dendritic DC2.4 cells (supplied by MSU, Moscow, Russia) were cultivated in the RPMI-1640 GlutaMAX-I medium (Servicebio) supplemented with 10% of fetal bovine serum (HiMedia) at 5% CO_2_ and 37 °C. The cells were subcultured every 2 days. At 24 h before transfection (for the HEK293T cell line) and 48 h before transfection (for the DC2.4 cell line), 15,000 cells per well were placed in 200 μL of the culture medium in a 96-well plate. The cells were next incubated at 37 °C and 5% CO_2_.

The cells in each well were transfected using the GenJect-40 transfection reagent (Molecta, Moscow, Russia) at the ratio of 0.25 μL of the reagent per 100 ng of mRNA. RNA was diluted to 100 ng/μL and employed at a rate of 10 ng per well. Before transfection, GenJect-40 was diluted with sterile 1× DPBS pH 7.5 (Servicebio) and incubated at room temperature for 10 min. After that, 25 μL of the diluted transfectant was added to each mRNA solution, mixed, and incubated for another 15 min at room temperature, and the mixture was added to the cells. Within each replicate, the transfected cells were incubated for 4, 24, or 48 h. The incubation conditions were the same as for routine culturing. For each construct, two biological and six technical replicates were set up at each time point.

### 2.5. An Assay of Luciferase Bioluminescence

To assess the luminescence of firefly luciferase, a 3× substrate solution (D-luciferin; Abisense, Sochi, Russia) was prepared, mixed with DPBS at a 1:2 ratio, and incubated at room temperature for 3 min. The cells were washed with 1× phosphate buffer pH 7.5, and 100 μL of the substrate was added to them. At this stage, cell lysis took place. Luminescence was immediately measured on a CLARIOstar Plus instrument (BMG Labtech, Ortenberg, Germany) at wavelengths of 100–560 nm (Gain = 3600).

To determine translation efficiency, the background bioluminescence level of an internal control containing no mRNA was subtracted from the obtained data. Total protein expression for each mRNA was calculated as cumulative luminescence corresponding to 2 days. Bioluminescence values are presented as the average of two independent measurements ± standard deviation; the data were normalized to the sample of mRNA capped with ARCA at a cap/GTP ratio of 4:1.

### 2.6. Loading of mRNA into LNPs

Encapsulation of mRNA into LNPs was performed according to a previously published protocol [19] with minor changes in lipid composition. For mRNA loading, 0.2 mg/mL mRNA in a buffer (10 mM citrate buffer pH 3.0) was used. The RNA was mixed with an ethanol solution of the lipid mixture in a microfluidic cartridge on a NanoAssemblr Benchtop instrument (Precision Nanosystems, Vancouver, BC, Canada). The five-component lipid mixture consisted of ionized lipidoid ALC-0315 (Sinopeg, Xiamen, Fujian, China), SM-102 (Sinopeg, Xiamen, China), DSPC (Avanti Polar Lipids, Alabaster, AL, USA), cholesterol (Merck Millipore, Burlington, MA, USA), and DMG-PEG-2000 (Avanti Polar Lipids, Alabaster, AL, USA) at a molar ratio (%) of 23.15:23.15:9.4:42.7:1.6. To form LNPs, the aqueous and ethanol phases were mixed in a ratio of 3:1 by volume; the total mixing rate was 10 mL/min.

After that, under sterile conditions, the nanoparticles were passed through a filter based on the PES 0.22 μm membrane (Merck, Rahway, NJ, USA) and stored at 4 °C. Storage time did not exceeded 3 days. Next, the quality of the obtained LNPs was analyzed using three parameters: particle size, the polydispersity index (PDI) (Zetasizer Ultra ZSP; Malvern Panalitycal, Westborough, MA, USA), and mRNA loading.

Analytical characterization of mRNA–LNP constructs included an assessment of RNA quality via capillary electrophoresis by means of MultiNA (Shimadzu). To disrupt the nanoparticles, we utilized detergent Triton X-100 (Sigma-Aldrich, Darmstadt, Germany). To this end, LNPs were mixed with 2 volumes of isopropanol and precipitated by centrifugation. The RNA pellet was washed with 75% EtOH and dissolved in deionized water free of RNases and DNases. The concentration of mRNA loaded into LNPs was determined by means of the difference in fluorescent signals [caused by staining with the RiboGreen reagent (Thermo Fisher Scientific)] before and after the disruption of LNPs.

The particle size was found to be 85.84 nm (RSD = 0.91), and the PDI did not exceed 0.14. The proportion of encapsulated RNA was 88.35%.

### 2.7. The Design of the In Vivo Experiment

Two- to four-month-old female C57BL/6JCit (B6) and I/StSnEgYCit (I/St) mice were bred and maintained under conventional, non-SPF conditions with water and food consumed ad libitum at Animal Facilities of the Central Tuberculosis Research Institute (Moscow, Russia).

B6 mice were used to evaluate the T- and B-cell immune responses that arise in animals after vaccination. These experiments involved a delayed-type hypersensitivity (DTH) test, an IFN-γ ELISPOT assay, and an IgG assay. Mice were randomly assigned to four groups: PBS (control), LNPs, mRNA–LNP, and BCG. In the first three groups, the administration procedure was the same: mice were immunized intramuscularly twice with an interval of 3 weeks. The dose of RNA was 50 μg per injection. The volume of PBS per dose was 100 μL (50 µL into each thigh). The dose of LNPs was equivalent to ±10% of the number of particles in the mRNA–LNP vaccine group. Three weeks after the second immunization, the mice were decapitated for subsequent analyses. In the case of immunization of mice with the BCG vaccine, single intramuscular administration was performed at a dose of 100,000 colony-forming units (CFUs) per animal at 5 weeks before decapitation.

I/St mice were utilized for the evaluation of protective immune responses because they are a tuberculosis-sensitive strain [20]. These animals were also randomly distributed into four groups. The immunization scheme for I/St mice was identical to that for B6 mice. Six weeks after the first immunization of mice of groups “PBS”, “LNPs”, and “mRNA–LNP” and at 5 weeks after immunization of mice of the BCG group, the animals were challenged with the virulent strain H37Rv of *M. tuberculosis* at a dose of 500,000 CFUs (intravenous infection). Some animals from each group (n = 4–5) were used to assess the bacterial load in the spleen and lungs, and the remaining animals from each group (n = 9–11) were monitored after the infection initiation to assess the dynamics of survival.

### 2.8. Bacterial Quantification

To determine vaccine efficacy, at 7 weeks after infection initiation, *M. tuberculosis* CFUs were counted in the lungs and spleen (4–5 mice per group). For this purpose, the lungs and spleens of infected mice were excised and homogenized in 2 mL of saline, and serial 10-fold dilutions of the organ homogenates were prepared and plated on Petri dishes with Middlebrook 7H10 agar at 50 μL per dish. Petri dishes with the applied suspensions were incubated at 37 °C. After 21 days, the number of macrocolonies of *M. tuberculosis* H37Rv in a dish was counted, and their number per organ was calculated.

### 2.9. Delayed-Type Hypersensitivity Test (DTH)

At 3 weeks after the second immunization of mice from groups PBS, LNPs, and mRNA–LNP and at 5 weeks after immunization of mice from the BCG group, the DTH test was performed (7–10 mice per group). The reaction was assessed as swelling of the left hind paw of a mouse in response to injection with 40 µL of PBS containing 50 IU of purified tuberculin (from the Scientific Centre for Expert Evaluation of Medicinal Products) at 48 h after the injection. The data are presented as ∆ (swelling of the left paw minus swelling of the right paw).

### 2.10. An Interferon-γ (IFN-γ) Enzyme-Linked Immunosorbent Spot (ELISPOT) Assay

The T-cell response in immunized mice was quantitated as the number of splenocytes secreting IFN-γ in response to treatment of the cells with a sonicate of (or full-length) mycobacterial protein ESAT6 (Rv3875), with the help of ELISPOT with kits Mouse IFNg ELISpot Set (BD Biosciences, Franklin Lakes, NJ, USA) and AEC Substrate Set (BD Biosciences) according to the manufacturer’s instructions. Rv3875 protein production and purification were performed according to a protocol described elsewhere [8]. The sonicate was a soluble fraction of the sonicated lysate of strain *M. tuberculosis* H37Rv and was represented by a set of pathogenic mycobacterial antigens. Splenocytes from the immunized mice’s spleens were isolated under sterile conditions by a procedure described before [21] and then were seeded at 200,000 cells/well in ELISpot plates with PVDF membrane (BD Biosciences), along with simultaneous stimulation with antigens. In the case of splenocyte treatment with both the sonicate and the ESAT6 protein, the concentration of the stimulatory agent in the final volume within a well of the multiwell plate was 10 μg/mL. Cells either without stimulation or stimulated with concanavalin A (Sigma-Aldrich, St. Louis, MO, USA) at a concentration of 5 μg/mL served as negative and positive controls, respectively. The number of spot-forming units (SFUs) corresponding to splenocytes secreting IFN-γ was determined.

### 2.11. Determination of IgG Titers to Antigens of M. tuberculosis

To assess the humoral response (titer of IgG antibodies to mycobacterial antigens), an enzyme immunoassay was performed by a method described previously [22]. To quantify antibodies, various serum dilutions from 1:50 to 1:400 were used.

### 2.12. Statistical Analysis

Because most of the data from both in vitro and in vivo analyses proved to be normally distributed, parametric ANOVA with the Bonferroni correction for multiple comparisons was chosen for statistical processing of the data. A two-way ANOVA was performed to analyze the effects of cap analog structure and of in vitro transcription conditions (molar ratio of the cap to GTP). Ratios 2:1 (ARCA and CapGG) and 0.4:1 (CapAG) were defined as “low concentration”, 4:1 (ARCA and CapGG) and 0.8:1 (CapAG) as “medium concentration”, and 8:1 (ARCA and CapGG) and 1:1 (CapAG) as “high concentration”.

A one-way ANOVA was applied to analyze the immune responses and bacterial load in mice. Overall survival was assessed by the Kaplan–Meier method. The significance of differences in overall survival was calculated by the Mantel–Cox logrank test. A *p* value above 0.05 was assumed to indicate the absence of a significant difference. Differences in the amounts of dsRNA and in total RNA yield were evaluated by the nonparametric Kruskal–Wallis test (Dunn’s multiple comparisons test). The nonparametric Spearman one-tailed test was chosen to examine the correlation between the amount of dsRNA and translation efficiency. Statistical calculations and graphical presentation of data were performed using GraphPad Prism software version 8.0.

## 3. Results

### 3.1. The Impact of the Cap Analogs on mRNA Translation

We evaluated effects of synthetic cap analog structure and of in vitro transcription conditions (ratio of a cap analog to GTP) on the translation of reporter mRNAs in two cell lines: HEK293T and DC2.4. Translation efficiency was determined by quantification of the bioluminescence of firefly luciferase (LucF).

We noticed that the cap type affected luciferase expression levels and bioluminescence in HEK293T cells [two-way ANOVA: 4 h: F(2, 93) = 141.8, *p* < 0.001; 24 h: F(2, 99) = 36.39, *p* < 0.001; 48 h: F(2, 99) = 21.81, *p* < 0.001] and in DC2.4 cells [two-way ANOVA: 4 h: F(2, 99) = 35.31, *p* < 0.001; 24 h: F(2, 99) = 64.91, *p* < 0.001; 48 h: F(2, 99) = 48.25, *p* < 0.001].

The results of comparison of mRNAs with inclusion of different cap types revealed that attachment of synthetic cap 1 analog CapGG or CapAG gave higher luciferase expression as compared to the mRNA capped with ARCA at most time points in both cell lines (Figure 2a,b). A comparison of the efficiency between the two cap 1 analogs indicated that bioluminescence levels after transfection of mRNA containing CapGG—under some conditions of in vitro transcription—were higher at 4 h post-transfection in HEK293T cells and at all time points in DC2.4 cells as compared to mRNA carrying CapAG.

Simple analysis of main effects suggested that the conditions of in vitro transcription (the ratio of a cap analog to GTP) did have a statistically significant effect on luciferase expression in HEK293T cells [two-way ANOVA: 4 h: F(2, 93) = 12.23, *p* < 0.001; 24 h: F(2, 99) = 1.76, *p* = 0.178; 48 h: F(2, 99) = 2.33, *p* = 0.103] and in DC2.4 cells [two-way ANOVA: 4 h: F(2, 99) = 3.59, *p* = 0.031; 24 h: F(2, 99) = 5.58, *p* = 0.005; 48 h: F(2, 99) = 7.88, *p* < 0.001].

A significant effect of in vitro transcription conditions on luciferase expression was observed for caps CapGG and ARCA in HEK293T cells at 4 h after transfection (Figure 2a). After transfection, mRNA synthesized with either CapGG or ARCA at a ratio of 4:1 yielded higher bioluminescence levels as compared to the mRNA transcribed at a ratio of 2:1 or 8:1. In cell line DC2.4, the mRNA capped with CapGG at a cap/GTP ratio of 4:1 was significantly different in bioluminescence intensity from the mRNA capped at a ratio of 8:1 at 24 h after transfection.

The lowest luciferase bioluminescence values were registered with mRNA carrying cap 0 (ARCA). Luciferase bioluminescence values for samples capped with this analog differed from the negative control (uncapped RNA) only at a ratio of 4:1 and only at time points “4 h” (HEK293T and DC2.4 cells) and “24 h” (DC2.4 cells) after transfection. In this context, samples synthesized at either low or high concentrations of this cap manifested the lowest levels of luciferase expression. We can hypothesize that a low ARCA/GTP ratio (2:1) and a high ARCA/GTP ratio (8:1) may lead to the formation of an elevated amount of impurities, whereas in the case of the 2:1 ratio, the capping efficiency may be inadequate. When carrying cap 0, the dsRNA admixture and uncapped RNA may be more effectively recognized by innate components of the immune system, and this phenomenon may cause translation suppression.

Thus, the results of our experiments on cultured cells indicate the best efficiency of the synthetic analog CapGG in terms of luciferase expression. The optimal setting for the synthesis of mRNA in this case is the ratio of the cap analog to GTP of 4:1.

### 3.2. Quality of mRNA and the mRNA–LNP Vaccine

At the next step, we estimated effects of the synthetic cap analogs’ structure and of the ratio of a cap analog to GTP on the quality of the obtained mRNA samples (Figure 3a–c). According to the results, there was no pronounced influence of cap structure on the dsRNA admixture amount or on the quantitative yield of RNA (Figure 3b,c). The concentration of dsRNA admixture in all studied samples did not exceed 3% and, according to a correlation analysis, did not significantly affect the intensity of luciferase bioluminescence at all time points (Appendix A).

Thus, there were no significant effects of capping conditions and of cap structure on the level of dsRNA and on the quality of RNA. Accordingly, for further work, synthetic cap analog m_2_^7,3′-O^GpppGmG “CapGG” was chosen for incorporation into mRNA vaccine mEpitope-ESAT6, as was the cap/GTP ratio of 4:1 (transcription conditions).

The vaccine mRNA was encapsulated into LNPs, and the quality of mEpitope-ESAT6 mRNA within a ready-for-use mRNA–LNP vaccine was analytically characterized, as were the LNPs themselves. The proportion of encapsulated RNA, RNA integrity, and characteristics of the LNPs (size and PDI) were acceptable (Figure 3d,e).

### 3.3. The Adaptive Immune Response in B6 Mice After the Vaccination

To evaluate the immunogenicity of the antituberculosis multiepitope mRNA vaccine mEpitope-ESAT6 carrying a new synthetic analog of the cap, titers of IgG antibodies against mycobacterial antigens were determined in B6 mice, as was the cellular response of splenocytes in terms of the number of IFN-γ producers after specific stimulation by the ELISpot method. The magnitude of the DTH reaction to tuberculin was quantified too (Figure 4a).

The vaccination caused a significant increase in titers of IgG antibodies against mycobacterial antigens at four dilutions: 1:50, 1:100, 1:200, and 1:400 [one-way ANOVA, 1:50: F(3, 15) = 67.58, *p* < 0.0001; 1:100: F(3, 15) = 33.83, *p* < 0.0001; 1:200: F(3, 15) = 28.02, *p* < 0.0001; 1:400: F(3, 15) = 52.39, *p* < 0.0001]. The mRNA–LNP vaccine and BCG induced a robust total IgG humoral immune response to mycobacterial antigens at all four dilutions of the serum significantly greater than the control PBS group achieved (Figure 4b). The IgG titer was higher in the mRNA–LNP group than in the BCG group across all four serum dilutions, with a significant difference observed at dilutions 1:50 and 1:200 (*p* = 0.006 and 0.011, respectively).

Furthermore, there was a significant effect of the vaccination on the DTH reaction to mycobacterial proteins (50 IU of purified tuberculin) and on ELISpot assay results as indicators of cellular immunity [one-way ANOVA, DTH: F(3, 31) = 11.98, *p* < 0.0001; ELISpot (ESAT6): F(3, 16) = 243469, *p* < 0.0001; ELISpot (a sonicate): F(3, 16) = 21.52, *p* < 0.0001].

Tuberculin-specific paw swelling in both vaccine groups was found to be significantly greater than in the control PBS group (*p* < 0.001 for both vaccinated groups) (Figure 4c). The magnitude of the DTH reaction in group mRNA–LNP (0.314 ± 0.063) was 8.4% higher as compared with the BCG group (0.290 ± 0.031) (without significance).

Stimulating splenocytes with the *M. tuberculosis* sonicate, we detected a significantly higher IFN-γ response in the mRNA–LNP vaccine group compared to the BCG group (*p* = 0.014) (Figure 4e). As a specific control, we also performed an ELISpot assay involving specific stimulation of splenocytes by the recombinant ESAT6 protein, whose epitopes are encoded in the mEpitope-ESAT6 vaccine (this protein is not expressed in the strains used for BCG production) [23]. The ELISpot assay revealed that, as expected, the frequency of ESAT6-specific IFN-γ-producing splenocytes was much higher after mRNA–LNP vaccination as compared to any other group, including BCG (Figure 4d).

Thus, our results suggest that the multiepitope mRNA vaccine mEpitope-ESAT6 leads to the formation of more pronounced humoral and cellular immunity as compared to BCG.

### 3.4. The Protective Response in I/St Mice After the Vaccination

The induction of protective immunity was investigated in the tuberculosis-sensitive I/St mouse strain (Figure 5a). Seven weeks after the *M. tuberculosis* challenge, mice were tested for bacterial load within the lungs and spleen as primary endpoints of protective efficacy. We noted a significant effect of the vaccination on bacterial load in the lungs and spleen [one-way ANOVA, lung: F(3, 15) = 5.76, *p* = 0.008; spleen: F(3, 16) = 6.44, *p* = 0.005]. Immunization of mice with the BCG vaccine-induced significant prophylactic protection as evidenced by a reduction in bacterial load in the spleen (Figure 5b) and a lung (Figure 5c) as compared to the PBS group (*p* = 0.006 and 0.04, respectively). The protective effect induced by the mRNA–LNP vaccine was clearly detectable but not statistically significant, causing a reduction by 0.308 log_10_ units in CFUs within a lung and a 0.535 log_10_ reduction in CFUs within the spleen.

Similar results were obtained in the survival experiment, which ended on day 136 post-challenge (Figure 5d). The Kaplan–Meier survival analysis uncovered a significant increase in the survival of BCG-vaccinated animals compared to the PBS group (Mantel–Cox logrank test, *p* < 0.001). In the group of mice immunized with mRNA–LNP, 8 out of 11 animals had died by the end of the experiment, whereas in the BCG group, only 3 out of 11 had died.

Taken together, these results indicate that despite a more pronounced adaptive response after immunization with the mRNA vaccine mEpitope-ESAT6 compared to BCG, the new vaccine does not provide effective protection after a challenge with *M. tuberculosis*.

## 4. Discussion

The creation of effective mRNA vaccines against bacterial pathogens, in particular *M. tuberculosis*, is a promising avenue of research in the fight against bacterial infections. In our study, we optimized a previously evaluated antituberculosis multiepitope mRNA vaccine based on ESAT6 [22]. For optimization, we tested various types of synthetic analogs of the mRNA cap. The results obtained with reporter mRNAs in HEK293T and DC2.4 cultured cells showed that the incorporation of cap 1 analog CapGG into reporter mRNA gives the best expression of luciferase in both cell lines. At the next stage, we assessed the benefit of using CapGG within the vaccine RNA and found that the administration of vaccine mEpitope-ESAT6 containing CapGG to mice induces cellular and humoral adaptive immunity. In this context, the immunogenicity of mEpitope-ESAT6 carrying CapGG was more pronounced as compared to vaccination with BCG. At the same time, mEpitope-ESAT6, unlike BCG, did not provide full protection of I/St mice from tuberculosis after an *M. tuberculosis* challenge; the new vaccine only slightly reduced mortality and bacterial load in the lungs and spleen of the animals. Altogether, our data indicate that the incorporation of a more effective cap analog into vaccine RNA can have a significant influence on the expression of cellular and humoral immune responses (by a factor of up to 4 as compared to the previous work on the same vaccine containing the ARCA cap [22]), but this effect did not provide protection from tuberculosis comparably to BCG.

An effective vaccine against tuberculosis should evoke cellular and humoral immunity to *M. tuberculosis* antigens. Furthermore, due to distinctive features of the pathogenesis of *M. tuberculosis* infection, both a CD8^+^ T-cell response and a CD4^+^ T-cell response are necessary in the initial phases of the infection; they can be successfully induced by ESAT6-based vaccines [24]. In addition, the vaccine should be able to form long-term immunity against mycobacterial antigens (immunological memory). In general, an effective antituberculosis vaccine should provide an optimal balance of Th1/Th2/Th17/Treg T-cell responses. In our vaccine, mEpitope-ESAT6, only epitopes of the secreted protein ESAT6 are used. ESAT6 is a key factor for the survival of *M. tuberculosis* inside macrophages and for its spread to other healthy cells. Through various signaling and metabolic pathways, ESAT6 participates in the inhibition of autophagy, in the triggering of necrosis, and in the stimulation of the production of cytokines IL-6, IL-1β, and IL-10 for the creation of a chronic proinflammatory background at a site of infection [25,26].

Sequences of highly immunogenic secretory proteins ESAT6, CFP10, and MPT64 are often employed in antituberculosis vaccines. *ESAT6* and *CFP10* are parts of the *RD1* region of difference between *M. tuberculosis* and BCG. Nonetheless, the most common antigens used in vaccines include secreted fibronectin-binding proteins Ag85A and Ag85B, which have mycolyltransferase activity, which is necessary for the maintenance of the integrity of the mycobacterial cell wall [5]. Existing ESAT6-containing subunit and vector vaccines, such as AEC/BC02 (Ag85B and ESAT6-CFP10), GamTBvac [Ag85A and ESAT6-CFP10], H56:IC31 [Ag85B, ESAT6, and Rv2660c], and H107e/CAF10b [PPE68, ESAT6, EspI, EspC, EspA, MPT64, MPT70, and MPT83] have shown high efficacy in various phases of clinical trials [23,27,28,29]. These results imply that ESAT6 is a key immunodominant antigen of *M. tuberculosis*. Nonetheless, our findings suggest that an mRNA vaccine encoding only ESAT6 epitopes does not provide protection from tuberculosis despite eliciting a robust adaptive immune response. A possible reason is that the adaptive response to ESAT6 alone may not be sufficient to ensure the effective elimination of this pathogen.

Examination of the results of four articles on the application of mRNA vaccines against tuberculosis indicates that they give rise to cellular and humoral adaptive immunity in mice (Table 1). Of the four papers analyzed, only two studies compared the effectiveness between mRNA vaccines and BCG [30,31]. Immunization with mRNA in these studies yielded more pronounced results on the formation of cellular and humoral immunity against mycobacterial antigens [30,31]. Nevertheless, consistent with our findings, mRNA vaccines did not provide pronounced unidirectional decreases in the bacterial load within the lungs and spleen. In all these papers, only individual organ- and dose-dependent effects of mRNA vaccination on bacterial load have been detected. An analysis of the impact of the vaccination on animal survival was not carried out in any of these studies.

In our previous project, we already assessed the efficacy of an mRNA vaccine encoding a multiepitope antigen based on the ESAT6 protein with the incorporation of cap 0: ARCA [22]. In the current study, we used a cap 1 synthetic analog called CapGG for cotranscriptional incorporation into the vaccine RNA, and this vaccine showed the best efficiency in the experiment on cultured cells. Meanwhile, other components of mRNA, the vaccination regimen, and the dose of the vaccine were not changed. In both studies, animals immunized with BCG served as the reference group. The comparison of the experimental results is presented in Table 2. We noticed that the mRNA vaccine mEpitope-ESAT6 carrying CapGG causes more pronounced humoral and cellular immune responses in mice than does the same mRNA carrying the ARCA cap or the BCG vaccine. On the other hand, the protective properties of our vaccine did not improve. Survival of mice in the mRNA–LNP group was 37.5% with the ARCA cap and 27.3% with the CapGG cap. Both types of mRNA only slightly reduced the bacterial load in the lungs and spleen, and these improvements were worse than those in the BCG group (both for the ARCA cap and for CapGG). It is worth clarifying that these two experiments were conducted at different time points, and therefore it is not entirely correct to compare the results directly.

A part of our work was aimed at finding optimal structural versions of the anti-reverse 5′ cap for inclusion into a vaccine. To this end, we synthesized mRNAs with cotranscriptional incorporation of a di- or trinucleotide cap analog at different molar ratios of the cap analog to GTP. The resultant mRNAs were analytically characterized and used to compare translation efficiency between two cell lines (HEK293T and DC2.4).

For mRNAs carrying analogs of cap 1, we observed higher overall protein expression as compared to the mRNA carrying cap 0 (Appendix A). This finding is consistent with data obtained previously on caps of different structures, including the class of anti-reverse cap analogs [34,35]. It is possible that the differences in translation efficiency between mRNAs carrying cap 0 or cap 1 are primarily due to differences in affinity (of signaling proteins and other effectors of innate immunity) for these structures. This is because 2′-O-methylation of the first transcribed nucleotide of mRNA is reported to have no significant influence on the susceptibility to decapping by hDcp2 and on affinity for eIF4E [36].

The cotranscriptional approach to cap 1 incorporation is utilized in other mRNA therapeutics [37,38,39]. Anti-reverse cap m_2_^7,3′-O^GpppAmG is used for capping in the approved anti-COVID-19 mRNA vaccine BNT162b2, developed by BioNTech [40]. Additionally, there are currently many mRNA therapeutics with post-transcriptional capping involving cap 1 analogs: antiviral vaccines mRNA-1273 [41], mRNA-1273.351, and mRNA-1273.211 [42]; an mRNA vaccine against Zika virus [43]; a vaccine against gastrointestinal cancer [44]; and gene replacement therapy [45,46]. In these cases, the cap of mRNA can have different proximal nucleotides.

## 5. Conclusions

Our data indicate that optimization of an RNA molecule via the incorporation of a synthetic analog of cap 1 of the m^7^GpppGmG type improves the expression of reporter proteins in cell lines and enhances the immunogenicity of the antituberculosis mRNA vaccine mEpitope-ESAT6 but does not affect the protective efficacy of the vaccine. To create a more effective mRNA vaccine against tuberculosis, further optimization of its antigen composition is likely required. It is necessary to include epitope sequences from a wider range of antigens as well as to improve the balance of epitope composition to induce an optimal ratio of a humoral response to a Th1/Th2/Th17/Treg T-cell response. Progress in silico modeling techniques can help to overcome these barriers. In addition, the use of antituberculosis mRNA vaccines for heterologous booster vaccination seems attractive. In particular, in the mRNA vaccine mEpitope-ESAT6, only sequences absent in BCG are used; this state of affairs also implies its possible application (and requires additional testing) as a booster vaccine.

## Figures and Tables

**Figure 1 vaccines-12-01267-f001:**
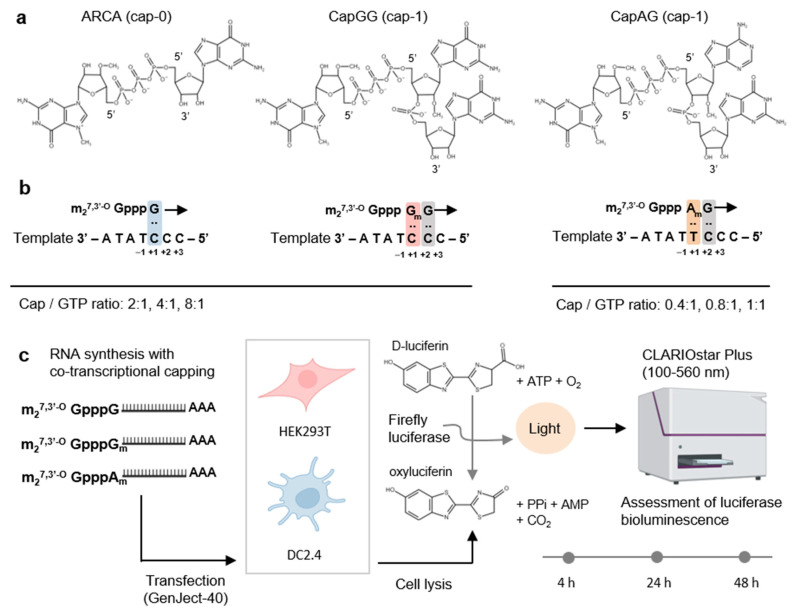
(**a**) Chemical structures of the trinucleotide 5′ cap analogs tested in this work: “CapAG” (m_2_^7,3′-O^GpppAmG), “CapGG” (m_2_^7,3′-O^GpppGmG), and dinucleotide cap “ARCA” (m_2_^7,3′-O^GpppG). The ARCA structure corresponds to cap 0, whereas the 2′-O-ribose–methylated cap analogs correspond to cap 1. (**b**) Incorporation of the dinucleotide cap is accompanied by annealing of guanosine to the +1 cytosine in the antisense strand of the template; the trinucleotide cap analogs can anneal at two sites (+1 and +2), thereby providing more efficient capping. To obtain mRNAs having different capping efficiency, three molar ratios of a cap analog to GTP were used; for ARCA and CapGG: 2:1, 4:1, and 8:1, and for CapAG: 0.4:1.0, 0.8:1.0, and 1:1. (**c**) The design of the experiment with reporter mRNAs.

**Figure 2 vaccines-12-01267-f002:**
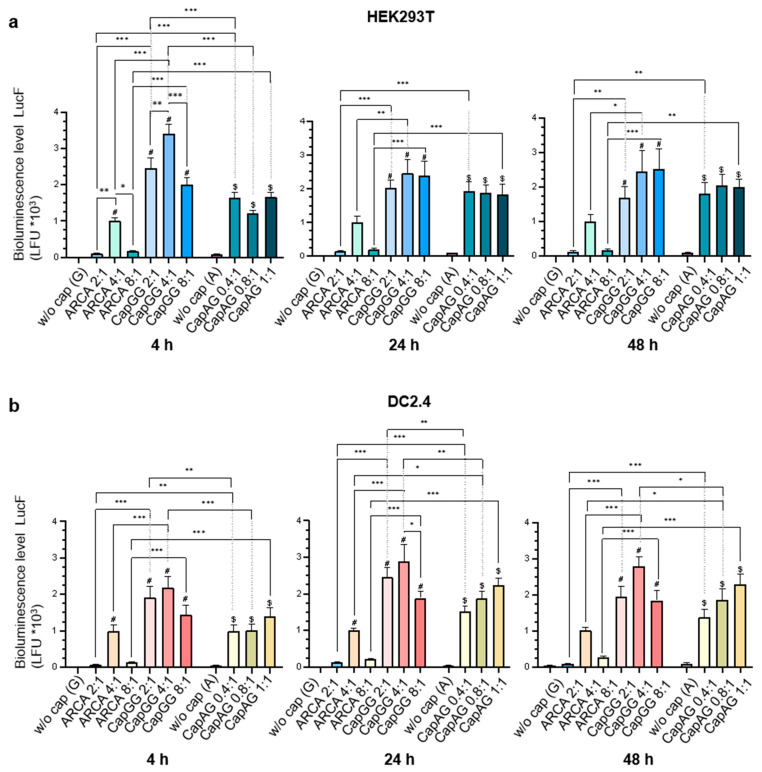
Relative intensity of luciferase bioluminescence in lysates of HEK293T (**a**) or DC2.4 (**b**) cells transfected with mRNAs carrying different cap analogs. The measurements were performed at 4, 24, and 48 h after transfection. The bars represent the mean ± SEM of two independent biological replicates, where each independent biological replicate consisted of six independent transfections; the means were normalized to ARCA-capped RNA. Statistical significance: * *p* < 0.05, ** *p* < 0.01, *** *p* < 0.001, (two-way ANOVA with Bonferroni’s adjustment); ^#^ a significant difference between the control and experimental samples for ARCA and CapGG groups; ^$^ for the CapAG group.

**Figure 3 vaccines-12-01267-f003:**
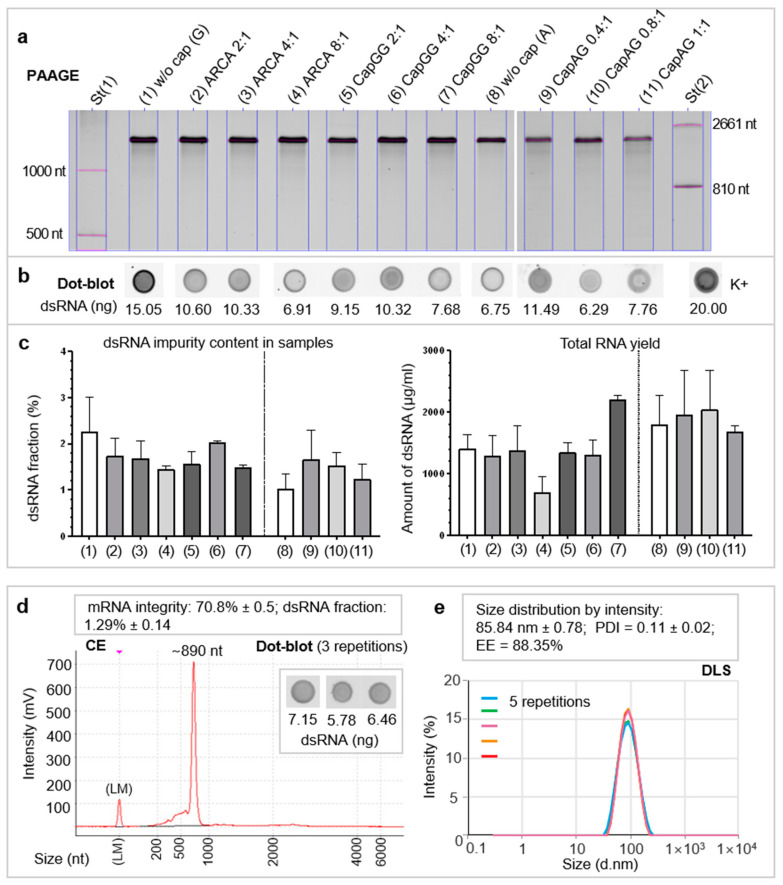
Analytical characterization of RNA samples. (**a**) Assessment of LucF mRNA quality by denaturing 6% polyacrylamide gel electrophoresis; (**b**) dsRNA was quantified by dot blot assay based on an anti-dsRNA IgG antibody and an anti-mouse IgG-PE antibody (500 ng of purified in vitro-transcribed RNA was subjected to the analysis). (**c**) Percentage of dsRNA impurities in mRNA samples and the yield of total RNA (μg/mL) in two in vitro transcription-independent transcription assays (two technical replicates for the dot blot assay and three technical replicates for RNA quantitation). Bars represent the mean (±SEM) normalized to ARCA-capped RNA. (**d**) Evaluation of mEpitope-ESAT6 mRNA integrity by capillary electrophoresis and estimation of the dsRNA content by the dot blot assay. Values are presented as a proportion (%) ± SD. (**e**) LNP size assessed by dynamic light scattering (DLS); the polydispersity index (PDI) and encapsulation efficiency (EE) of mRNA into LNPs are given below. Values are given as a proportion (%) ± SD.

**Figure 4 vaccines-12-01267-f004:**
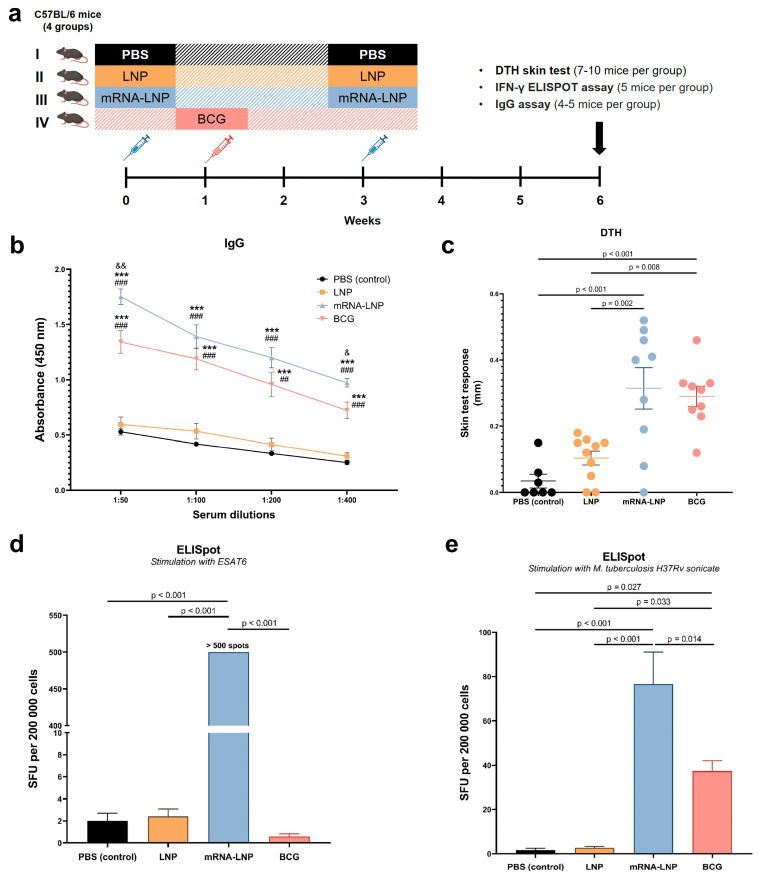
T- and B-cell immune responses after vaccination of C57BL/6JCit mice. Groups I, II, and III received two injections, three weeks apart: PBS buffer for Group I, LNP for Group II, and mRNA-LNP for Group III. Group IV received a single injection of BCG. (**a**) Experimental design of the in vivo experiments. (**b**) Plasma IgG antibody binding to mycobacterial antigens. Serum dilutions of 1:50 to 1:400 were used. (**c**) The DTH reaction: differences (∆) between the right and left paw at 48 h after injection of 50 IU of tuberculin. The IFN-γ ELISpot response from vaccinated mice stimulated with (**d**) ESAT6 or (**e**) the *M. tuberculosis* sonicate. SFU: spot-forming units. The data are presented as mean ± standard error. *p* values in panels (**b**–**e**) were determined by one-way ANOVA with Bonferroni’s post hoc test. *** *p* < 0.001 as compared with the PBS group; ^##^ *p* < 0.01, ^###^ *p* < 0.001 as compared with the LNP group; ^&^ *p* < 0.05 and ^&&^ *p* < 0.01 as compared with the BCG group.

**Figure 5 vaccines-12-01267-f005:**
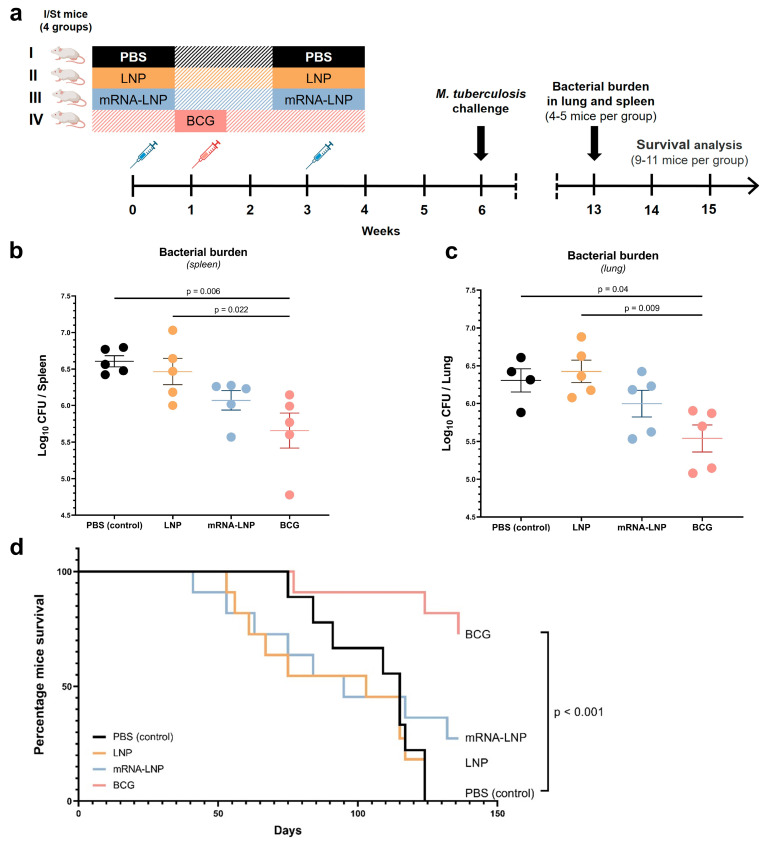
The evaluation of protective immune responses after vaccination of I/St mice. Groups I, II, and III received two injections, three weeks apart: PBS buffer for Group I, LNP for Group II, and mRNA-LNP for Group III. Group IV received a single injection of BCG. (**a**) Experimental design. Bacterial load assessed as CFUs in lung (**b**) and spleen (**c**) homogenates at 7 weeks post-challenge; (**d**) Kaplan–Meier survival curves. The data are presented as mean ± standard error. *p* values in panels b and c were determined by one-way ANOVA with Bonferroni’s post hoc test.

**Table 1 vaccines-12-01267-t001:** An overview of mRNA vaccines against *M. tuberculosis*.

Name/ Type/ Delivery	Antigens	Immunization Protocol	Results	Ref.
ID91-RNA (repRNA) NLC	Rv3619 (esxV; ESAT6-like protein), Rv2389 (RpfD), Rv3478 (PPE60), Rv1886 (Ag85B)	Intramuscularly, 1 µg (single or double immunization) Protective effect: 3 weeks after a challenge with a low-dose aerosol (LDA 50–100 CFUs) of *M. tuberculosis* H37Rv	*Single* Humoral: Yes Cellular: Yes Protective: Lungs: Yes Spleen: No *Double* Humoral: Yes Cellular: Yes Protective: Lungs: No	[32]
mRNA^CV2^ mRNA LNP	Ag85B CysD	intramuscularly, 5 µg of mRNA 3 times at 3-week intervals Protective effect: 5 weeks after final vaccination, mice were challenged with *M. tuberculosis* H37Rv (~100 viable bacilli per mouse)	Humoral: **Yes** Cellular: **Yes** Protective: Lungs: Yes Spleen: No	[30]
mRNA-Hsp65 mRNA naked	Hsp65 protein from *M. leprae*	intranasal single dose of 5 or 10 μg of mRNA Intranasal challenge was implemented by administration of 10^5^ viable CFUs of *M. tuberculosis* H37Rv 30 days after immunization	5 μg Humoral: n.a. Cellular: No Protective: No 10 μg Humoral: n.a. Cellular: Yes Protective: Yes	[33]
Sin.83 RNA naked	MTP83	Intramuscularly, 50 μg of RNA on four occasions at 3-week intervals Infection was induced by injecting 5 × 10^5^ viable CFUs of *M. tuberculosis* H37Rv 4 weeks or 6 months after the last RNA injection	Humoral: **Yes** Cellular: **Yes** Protective: Yes	[31]

NLC: nanostructured lipid carrier. n.a.: not assessed. Vaccines that showed better effectiveness as compared to BCG are highlighted in bold.

**Table 2 vaccines-12-01267-t002:** The comparison of efficacy between mRNA vaccines with different cap types (ARCA versus CapGG) prepared in this study and in our previous work [22].

Test	ARCA-TPL-mEpitope-ESAT6-MOD [22]		CapGG-TPL-mEpitope-ESAT6-MOD [Current Study]	
	BCG	mRNA–LNP	BCG	mRNA–LNP
ELISpot (stimulation with a sonicate)	++ 33.3 ± 8.4 spots	+ 12.8 ± 1.9 spots, 62% lower than in the BCG group	++ 37.4 ± 4.7 spots	+++ 76.6 ± 14.5 spots, 105% higher than in the BCG group
DTH test	++ 0.25 ± 0.04 mm	+ 0.21 ± 0.05 mm, 16% lower than in the BCG group	++ 0.29 ± 0.03 mm	+++ 0.31 ± 0.06 mm, 7% higher than in the BCG group
IgG assay (serum dilution of 1:200)	++ Absorbance (450 nm) = 0.39 ± 0.06	+ Absorbance (450 nm) = 0.37 ± 0.02, 5% lower than in the BCG group	++ Absorbance (450 nm) = 0.96 ± 0.11	+++ Absorbance (450 nm) = 1.20 ± 0.09, 25% higher than in the BCG group
Bacterial load in lungs (CFUs)	++ 83% reduction compared to the PBS group	+ 80% reduction compared to the PBS group	++ 81% reduction compared to the PBS group	+ 44% reduction compared to the PBS group
Bacterial load in spleen (CFUs)	++ 88% reduction compared to the PBS group	+ 65% reduction compared to the PBS group	++ 84% reduction compared to the PBS group	+ 68% reduction compared to the PBS group
Survival analysis	++ Survival rate 100%	+ Survival rate 37.5%	++ Survival rate 72.7%	+ Survival rate 27.3%

The magnitude of a response is denoted by plus signs, where “+” is a weak response, “++” denotes a moderate response, and “+++” is a strong response.

## Data Availability

Raw data from this study are available upon reasonable request.

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
