# Peer review of "A Cap-Optimized mRNA Encoding Multiepitope Antigen ESAT6 Induces Robust Cellular and Humoral Immune Responses Against Mycobacterium tuberculosis"

_vaccines, 2024, doi:10.3390/vaccines12111267_

Round 1
Reviewer 1 Report
Comments and Suggestions for Authors
Kozlova et al. reported an mRNA vaccine for the prevention of TB disease. Their results showed higher cellular and humoral responses, but it didn't protect the animals better than BCG. Recent reports of mRNA vaccines have attracted researchers because of the success of COVID-19 mRNA vaccines. Similarly, the authors approached the mRNA vaccine format for TB using a cap-optimized mRNA encoding multiepitope antigen. The study is interesting, and although the protection was not more effective than BCG, it provides valuable information for developing future vaccines by improving the design.
Why don't the authors check the efficacy in diversity outbred mice ?
Why the authors challenge with IV route? May be aerosol challenge will be effective!
Does the author performed histology studies ?
Author Response
Dear Editor and Referees:
Thank you for allowing us to submit a revised draft of our manuscript titled “A Cap-Optimized mRNA Encoding Multiepitope Antigen ESAT-6 Induces Robust Cellular and Humoral Immune Responses against Mycobacterium Tuberculosis”. We are grateful for your review of the manuscript and your valuable comments and concerns. We have been able to incorporate into the manuscript most of the suggestions provided by the reviewers.
We have highlighted the revisions within the manuscript.
Below, marked in red, are point-by-point responses to the reviewers’ comments and concerns.
Sincerely,
Vasiliy Reshetnikov
Reviewer 1
Comments and Suggestions for Authors
Kozlova et al. reported an mRNA vaccine for the prevention of TB disease. Their results showed higher cellular and humoral responses, but it didn't protect the animals better than BCG. Recent reports of mRNA vaccines have attracted researchers because of the success of COVID-19 mRNA vaccines. Similarly, the authors approached the mRNA vaccine format for TB using a cap-optimized mRNA encoding multiepitope antigen. The study is interesting, and although the protection was not more effective than BCG, it provides valuable information for developing future vaccines by improving the design.
Why don't the authors check the efficacy in diversity outbred mice?
Reply: Thank you for pointing this out. Indeed, we chose a tuberculosis-sensitive inbred line for evaluation of protective immune responses (doi:10.1016/j.tube.2004.09.014). This allowed us to evaluate the protective efficacy of the mRNA vaccine in a shorter time frame, since disease development in the insensitive line can take much longer. Nevertheless, the use of outbred mice could indeed provide a more complete picture of the immune response and vaccine efficacy in a genetically heterogeneous population. We consider the possibility of checking the efficacy in diversity outbred mice in the future when optimizing the vaccine design.
Susceptibility to TB infection is genetically restricted. In fact, only 5-10% of infected individuals develop clinical forms of TB. Accordingly, these susceptible 5-10% are the focus group for anti-TB vaccination. Most mouse strains and outbred mice are resistant to TB infection. Therefore, we chose the susceptible strain as the most relevant for our experiments.
Why the authors challenge with IV route? May be aerosol challenge will be effective!
Reply: We chose the IV route because it is more effective in delivering a high dose of M. tuberculosis inoculum (doi: 10.1128/AAC.00595-10). Also, this method allows more precise control of the dose of infection.
Does the author performed histology studies?
Reply: Thank you for your question. Unfortunately, we did not perform histology studies, as it was beyond the scope of the current study.

Reviewer 2 Report
Comments and Suggestions for Authors
Vasiliy Reshetnikov's team studied “A Cap-Optimized mRNA Encoding Multiepitope Antigen
ESAT6 Induces Robust Cellular and Humoral Immune Responses against Mycobacterium Tuberculosis”.
Among the different infectious diseases, tuberculosis is one of the most deadly infectious diseases. Every year millions of people are getting infected with Mycobacterium tuberculosis, though effective therapeutics are available, there is no notable vaccine in use. The authors have considered the ESAT6 antigen to produce mRNA vaccine. In addition, the authors have also used the different ratios of cap analogues of different structures to GTP. During this titration, the authors measured the formation of dsRNA, which is a by-product of the IVT mRNA preparation. Overall the study is well done.
Here are the comments from this reviewer:
What are 5UTR and 3UTR for the cassette?
What is TPL means?
Line 147: “cultured at 30 C and 180 rpm” are you sure? This is not normal conditions for bacterial cultures. Any explanation?
Line 177: How the membrane was dried.
Line 119-120: Please explain clearly; what you mean by removing ssRNA by nuclease S1.
Section 2.6: The molar ratios should be clearly explained in terms of weight ratios and how the stocks are prepared followed by dilution.
- LNPs stored at 4 degrees? How long?
Line 276: 50 ug is too much. COVID-19 vaccines for humans are 30 ug. How do you justify that?
Line 277: How 100 ul was administered by IM, to one thigh or two.
Section 2.9: For ELISpot assay we generally use a peptide pool. How the authors achieved peptide-specific T cell with whole protein.
Author Response
Dear Editor and Referees:
Thank you for allowing us to submit a revised draft of our manuscript titled “A Cap-Optimized mRNA Encoding Multiepitope Antigen ESAT-6 Induces Robust Cellular and Humoral Immune Responses against Mycobacterium Tuberculosis”. We are grateful for your review of the manuscript and your valuable comments and concerns. We have been able to incorporate into the manuscript most of the suggestions provided by the reviewers.
We have highlighted the revisions within the manuscript.
Below, marked in red, are point-by-point responses to the reviewers’ comments and concerns.
Sincerely,
Vasiliy Reshetnikov
Reviewer 2
Vasiliy Reshetnikov's team studied “A Cap-Optimized mRNA Encoding Multiepitope Antigen
ESAT6 Induces Robust Cellular and Humoral Immune Responses against Mycobacterium Tuberculosis”.
Among the different infectious diseases, tuberculosis is one of the most deadly infectious diseases. Every year millions of people are getting infected with Mycobacterium tuberculosis, though effective therapeutics are available, there is no notable vaccine in use. The authors have considered the ESAT6 antigen to produce mRNA vaccine. In addition, the authors have also used the different ratios of cap analogues of different structures to GTP. During this titration, the authors measured the formation of dsRNA, which is a by-product of the IVT mRNA preparation. Overall the study is well done.
Here are the comments from this reviewer:
What are 5UTR and 3UTR for the cassette?
Reply:
In the in vitro experiments, we used a cassette containing the 5'UTR and 3'UTR from the mRNA vaccine mRNA-1273 (Moderna, Cambridge, MA, USA), as specified in the Materials and Methods section 2.2 (lines 141-144). In the in vivo experiment, we used a cassette containing the 5'UTR, referred to as TPL, and the 3'UTR from mRNA-1273 (lines 144-145). The nucleotide sequence of this cassette is provided in the supplementary materials (Sequence S1).
What is TPL means?
Reply:
TPL (tripartite leader) is a nucleotide sequence in the 5′UTR of adenovirus late mRNAs. In our previous study, we demonstrated that this 5′UTR within mRNA constructs exhibits high translational potential in both in vitro and in vivo experiments (10.3390/ijms25020888).
Line 147: “cultured at 30 C and 180 rpm” are you sure? This is not normal conditions for bacterial cultures. Any explanation?
Reply:
Thank you for your question. You are correct; these conditions may appear unusual for standard bacterial cultures. However, NEB-stable cells are specifically recommended to be cultured at 30 °C rather than the typical 37 °C due to their sensitivity to higher temperatures, which can improve plasmid stability and overall yield in this context.
Line 177: How the membrane was dried.
Reply: Thank you for the remark. After the application, the membrane was air-dried at room temperature. Text has been corrected.
Line 119-120: Please explain clearly; what you mean by removing ssRNA by nuclease S1.
Reply: We used S1 nuclease during the preparation of the dsRNA standard to remove any single-stranded RNA (ssRNA) that remained unhybridized (lines 197–199). This step allows us to obtain a clean dsRNA mixture, which is essential for constructing the calibration curve and for subsequent calculations (Figure S1).
Section 2.6: The molar ratios should be clearly explained in terms of weight ratios and how the stocks are prepared followed by dilution.
Reply: The volumes of stock solutions were calculated based on the molecular weights of the components to achieve the required molar ratios, taking into account the mass of RNA used for formylation. Stock solutions were prepared at high concentrations and then the components were mixed in the given molar ratios by volume. For 2100 µl of the mixture, the following volumes were added: 105.00 µl of stock solution ALC-0315 (766.30 g/mol) with a concentration of 40.0 mg/mL, 97.23 µl of stock solution SM-102 (710.18 g/mol) with a concentration of 40.0 mg/mL, 175. 77 μl from stock solution of DSPC (790.15 g/mol) with a concentration of 10.0 mg/mL, 390.60 μl from stock solution of Cholesterol (386.66 g/mol) with a concentration of 10.0 mg/mL, 94.76 μl from stock solution of DMG-PEG 2000 (7509.20 g/mol) with a concentration of 10.0 mg/mL. The solution was perfused with 96% ethanol (1236.64 μL) to the final volume. This information seems excessive for the manuscript, so we did not include it in the text.
LNPs stored at 4 degrees? How long?
Reply: LNPs stored at 4 degrees for up to 3 days. We have supplemented the text of the relevant section (lines 251-252).
Line 276: 50 ug is too much. COVID-19 vaccines for humans are 30 ug. How do you justify that?
Reply: Thank you very much for raising this important question. Indeed, a single dose of the COVID-19 vaccine BNT162b2 (Pfizer) contains 30 µg of RNA, while another COVID-19 vaccine, mRNA-1273 (Moderna), contains 100 µg of RNA per dose. The 50 µg dose administered to mice in our study is indeed comparable to a human dose. However, as our own experience and that of other researchers in mRNA vaccine development shows, this human-equivalent dose is essential to elicit a quantitatively detectable immune response in mice. Comparable mRNA doses targeting bacterial infections administered in mice can be found in various studies: 50 µg of an mRNA vaccine against M. tuberculosis (10.1128/IAI.72.11.6324-6329.2004), 10 µg of an mRNA vaccine against B. burgdorferi (10.1038/s41541-024-00890-4), 15 µg of an mRNA vaccine against S. pyogenes (10.1016/j.vaccine.2016.11.040), among others. On average, mRNA doses in mouse studies range from 5 to 50 µg, depending on various parameters, including the presence/absence of modified nucleotides in the RNA, the packaging and delivery method, and the type of RNA (linear, circular, self-amplifying), among other factors.
Line 277: How 100 ul was administered by IM, to one thigh or two.
Reply:
The 100 µl volume was administered intramuscularly by injecting 50 µl into each thigh. We have clarified this in the revised version of our manuscript (line 278).
Section 2.9: For ELISpot assay we generally use a peptide pool. How the authors achieved peptide-specific T cell with whole protein.
Reply: Thank you for pointing this out. In our ELISpot assay, we stimulated splenocytes with the full-length mycobacterial protein ESAT6 (Rv3875) as well as a sonicate of M. tuberculosis H37Rv. Regarding the first stimulation method using the full-length protein, this is a standard approach in such studies. Splenocytes represent a heterogeneous population of cells isolated from the spleen. In addition to T cells, this population includes dendritic cells, which can process and present antigens to T cells during the ELISpot assay. This enables us to measure T-cell responses even when using full-length protein, as dendritic cells process the protein into peptides and present them to T cells, allowing for the detection of antigen-specific T-cell activation.

Reviewer 3 Report
Comments and Suggestions for Authors
A Cap-Optimized mRNA Encoding Multiepitope Antigen ESAT6 Induces Robust Cellular and Humoral Immune Responses against Mycobacterium Tuberculosis.
vaccines-3284003
The authors have investigated the effects of cotranscriptional capping conditions and of cap structure on the magnitude of the mRNAs’ translation in HEK293T and DC2.4 cells. The most effective cap version created an antituberculosis mRNA vaccine called mEpitope-ESAT6. We compared the immunogenicity and protective activity between mEpitope-ESAT6 and BCG and found that the vaccine with the new cap type is more immunogenic than BCG. Nonetheless, the increased immunogenicity did not enhance vaccine-induced protection. Thus, the incorporation of different cap analogs into mRNA allows for the modulation of the efficacy of mRNA vaccines.
Overall, the study and the article are extremely well-designed and well-executed. This can become a potential new vaccine for BCG. The methods section is described in great detail.
The discussion section is also extremely well written. It is highly interesting to read this study.
This article can be published with minor revisions
1. Please check the font color for Line 203 "two" word.
Author Response
Dear Editor and Referees:
Thank you for allowing us to submit a revised draft of our manuscript titled “A Cap-Optimized mRNA Encoding Multiepitope Antigen ESAT-6 Induces Robust Cellular and Humoral Immune Responses against Mycobacterium Tuberculosis”. We are grateful for your review of the manuscript and your valuable comments and concerns. We have been able to incorporate into the manuscript most of the suggestions provided by the reviewers.
We have highlighted the revisions within the manuscript.
Below, marked in red, are point-by-point responses to the reviewers’ comments and concerns.
Sincerely,
Vasiliy Reshetnikov
Reviewer 3
A Cap-Optimized mRNA Encoding Multiepitope Antigen ESAT6 Induces Robust Cellular and Humoral Immune Responses against Mycobacterium Tuberculosis.
vaccines-3284003
The authors have investigated the effects of cotranscriptional capping conditions and of cap structure on the magnitude of the mRNAs’ translation in HEK293T and DC2.4 cells. The most effective cap version created an antituberculosis mRNA vaccine called mEpitope-ESAT6. We compared the immunogenicity and protective activity between mEpitope-ESAT6 and BCG and found that the vaccine with the new cap type is more immunogenic than BCG. Nonetheless, the increased immunogenicity did not enhance vaccine-induced protection. Thus, the incorporation of different cap analogs into mRNA allows for the modulation of the efficacy of mRNA vaccines.
Overall, the study and the article are extremely well-designed and well-executed. This can become a potential new vaccine for BCG. The methods section is described in great detail.
The discussion section is also extremely well written. It is highly interesting to read this study.
This article can be published with minor revisions
- Please check the font color for Line 203 "two" word.
Reply: Thank you, the text color for line 203 "two" word has been corrected.
